# Comparative Analysis of Supervised Classifiers for the Evaluation of Sarcopenia Using a sEMG-Based Platform

**DOI:** 10.3390/s22072721

**Published:** 2022-04-01

**Authors:** Alessandro Leone, Gabriele Rescio, Andrea Manni, Pietro Siciliano, Andrea Caroppo

**Affiliations:** National Research Council of Italy, Institute for Microelectronics and Microsystems, 73100 Lecce, Italy; gabriele.rescio@cnr.it (G.R.); andrea.manni@le.imm.cnr.it (A.M.); pietro.siciliano@le.imm.cnr.it (P.S.); andrea.caroppo@cnr.it (A.C.)

**Keywords:** sarcopenia, surface EMG, machine learning, ageing

## Abstract

Sarcopenia is a geriatric condition characterized by a loss of strength and muscle mass, with a high impact on health status, functional independence and quality of life in older adults. To reduce the effects of the disease, just the diagnostic is not enough, it is necessary more than recognition. Surface electromyography is becoming increasingly relevant for the prevention and diagnosis of sarcopenia, also due to a wide diffusion of smart and minimally invasive wearable devices suitable for electromyographic monitoring. The purpose of this work is manifold. The first aim is the design and implementation of a hardware/software platform. It is based on the elaboration of surface electromyographic signals extracted from the Gastrocnemius Lateralis and Tibialis Anterior muscles, useful to analyze the strength of the muscles with the purpose of distinguishing three different “confidence” levels of sarcopenia. The second aim is to compare the efficiency of state of the art supervised classifiers in the evaluation of sarcopenia. The experimentation stage was performed on an “augmented” dataset starting from data acquired from 32 patients. The latter were distributed in an unbalanced manner on 3 “confidence” levels of sarcopenia. The obtained results in terms of classification accuracy demonstrated the ability of the proposed platform to distinguish different sarcopenia “confidence” levels, with highest accuracy value given by Support Vector Machine classifier, outperforming the other classifiers by an average of 7.7%.

## 1. Introduction

According to the United Nations report published in 2019 [1], the largest growth in the world population in terms of numerical terms is occurring in the age group 65 and over.

Aging is a multifactorial process that is associated with numerous changes in body composition, including bone mass, muscle mass and adipose tissue composition. Muscle, being the largest organ in the body constituting 40% of body mass, shows a progressive reduction in size and number of muscle fibers (up to 30%) in an age-dependent manner from 25 to 80 years of age [2]. This loss of muscle mass and strength may result in “sarcopenia”.

The term “sarcopenia” was first introduced by Rosenberg [3] and it is derived from the Greek words, “sarx” meaning flesh and “penia” meaning loss. This type of disorder is mainly observed in advanced stages in the elderly, but it is known that individuals start losing muscle from about 40 years of age or even younger and the rate of muscle loss accelerates with age [4]. In addition, the study reported in [5] shows that about 5–13% of people aged 60–70 years are sarcopenic, and among people aged 80 years and older, the prevalence has been estimated to be up to 50%. Although there is little data on the economic burden of this type of disease, its impact on national health services and hospitalized individuals is expected to be high [6]. In a study [7] published in 2004 and conducted in America on a sample of adults over 60 years of age, it was reported that the estimated health care cost attributable to sarcopenia was $18.5 billion ($10.8 billion in men, $7.7 billion in women). Furthermore, in more recent studies published in 2013 and 2015, an increase in costs from sarcopenia observed through computed tomography was shown [8,9]. Sarcopenia is not easily diagnosed. Moreover, the treatment of sarcopenia is still challenging because it is not easy to evaluate the time course of its three basic components, which are (1) muscle mass, (2) physical performance (such as walking speed), and (3) muscle strength. Many different methods have been adopted for the prevention of the disease, but exercise and physical activity are considered the most effective [10].

Currently, both muscle mass and strength are assessed by various invasive gold standard techniques such as Magnetic Resonance Imaging (MRI), Computed Tomography (CT) or Dual Energy X-ray Absorptiometry (DXA) [11,12,13]. These clinical examinations, although widely used, seem to be poorly utilized in this setting due to high equipment costs and lack of portability. In addition, the equipment/devices involved in these types of clinical examinations require highly trained medical personnel.

To overcome this limitation and extend the number of devices useful for sarcopenia assessment, smart sensors are becoming increasingly popular in recent years. For example, wearable devices, mobile apps, and embedded systems are frequently considered in healthcare and their use can undoubtedly be of support in providing early diagnosis and monitoring of sarcopenia patients.

Electromyography (EMG) is a reliable method of monitoring muscle fatigue and assessing the function and efficiency of muscles, which is done by identifying their electrical potentials. There are two types of electromyography analysis: the superficial measurement method (sEMG) and the intramuscular technique. The first one is less invasive since it uses non-invasive surface electrodes on the skin [14].

Several papers in the literature have focused on the use of EMG signals in the medical context [15,16,17,18,19]. In [17], for example, the potential clinical value of sEMG-based techniques in rehabilitation medicine with a focus on neurorehabilitation has been addressed. Yu et al. [18] developed a wireless medical sensor measurement system, including EMG, motion detection and muscle strength, to detect fatigue in multiple sclerosis patients.

In addition, the research activity on fatigue/muscle strength analysis is continuously evolving. In [20] the authors calculated the muscle strength of the biceps brachii using trained and untrained subjects in static contraction (isometric contraction). This work analyses the time-frequency response of muscle contraction obtained from the analysis of raw signals, captured by bipolar surface electrodes placed on the belly of the biceps brachii. In [21] an EMG patch was designed and developed, which could be worn on the lower leg (gastrocnemius muscle), to detect muscle fatigue in real-time during exercise. They also designed and developed an app to display muscle fatigue levels and end-user riding information. In this context, good performance was obtained using supervised Machine Learning (ML) schemes. The study reported in [22] investigated the disorder of atrophy by analyzing the recorded sEMG signals, considering the biceps for detecting atrophy and evaluating three different classifiers (Linear Discriminant Analysis, Quadratic Discriminant Analysis and Support Vector Machine—SVM) to separate the samples into two atrophic and normal classes using different sets of extracted features. The results showed that Quadratic Discriminant Analysis was the most suitable classifier for detecting this specific disorder.

In [23] the authors introduced two new architectures based on Recurrent Neural Networks to overcome the difficulties typically encountered in real-time classification of EMG signals. The performance of these architectures outperformed a number of state-of-the-art methods by achieving a classification accuracy of 96% while reducing the delay time to the value of 600 ms. In addition, the work discussed in [24] investigated the feasibility of a specific ML classifier (SVM) for identifying upper limb intention from sEMG signals by developing a novel human-machine interface for self-rehabilitation training of stroke patients.

As highlighted by the state of the art introduced above, sEMG is widely used for the analysis of specific pathologies but very few works in the literature have focused on the use of sEMG for the assessment of sarcopenia or its monitoring over time. For example, the study proposed in [25] compared the muscle strength of seventy-one hip fracture patients based on the presence of sarcopenia after surgery and correlated the measured values between sEMG and dynamometer in the postoperative measurement of muscle strength. After the trial, the authors concluded that dynamometer and sEMG values were highly correlated, although no statistically significant difference in muscle strength with or without sarcopenia was evidenced. On the other hand, the primary purpose of the research reported in [26] was to examine whether age-specific effects could be observed in the sEMG representation of healthy individuals engaged in cyclic back extension exercise, seeking to develop new biomarkers that could be used to screen for very early forms of sarcopenia.

Motivated by the goal of assessing sarcopenia, the primary objective of this research is the design and implementation of a new platform that integrates smart sEMG technology and a software module as a Decision Support System (DSS) for healthcare personnel. The developed system can provide additional information useful for assessing the user’s muscle condition during physical performance evaluation (typically performed via sit-to-stand testing and gait speed) in a cost-effective and non-intrusive manner. In addition, a minimally invasive and easy-to-use DSS that can also be used in nursing homes (or at home) may facilitate more frequent monitoring than hospital-based surveys. Thus, it can provide risk indices for sarcopenia or muscle decay, even in the prevention phase. Then, more in-depth, and invasive, but more accurate, analysis may be required in the clinical setting. Finally, this type of DSS can also monitor disease progress. In this study, ML, a sub-branch of artificial intelligence that allows a model to learn automatically from data, was used. Specifically, the efficiency of eight ML classifiers that enjoy great popularity in this research area was compared, as there is no predefined and validated model that guarantees good performance with any type of test data. The integrated platform was included in an initial validation in a hospital ward to demonstrate the effectiveness of the platform.

## 2. Materials and Methods

The sEMG-based platform detailed in this work is a DSS for specialized medical personnel as well as caregivers, since the entire system is user-friendly in terms of its functionality and output analysis. The proposed platform is made up of two main components: a hardware system for collecting sEMG signals and a software tool for processing raw EMG data and extracting sarcopenia-related features. Figure 1 shows a high-level overview of the platform.

Figure 2 shows the positioning of the electrodes on the muscles of the leg involved in the experimentation (left side) and the interface of the acquisition software with the highlighted waveforms of the signals acquired (right side).

Wearable sEMG probes are integrated into the proposed system for raw signals collection. A commercial product was employed in the final version of the platform. The employed probes are part of the FREEEMG1000 system, produced by the BTS Bioengineering situated in Garbagnate Milanese, Italy [27]. In addition to the hardware platform, BTS Bioengineering provides software libraries for developers (written in the C# programming language) that also consent to full access to the raw data collected by the probes. The system is entirely based on wireless technology, and it can use up to ten lightweight, minimally invasive wireless EMG probes (dimensions are 41.5 × 24.8 × 14 mm and the weight is about 13 g). The probes are clipped to the pre-gelled Silver/Silver Chloride (Ag/AgCl) electrodes, providing for a rapid, simple and stable mounting for the user’s movements at the highest level of usability. The active electrodes permit us to amplify the signals, digitize them on board and communicate with a USB receiver connected directly to an elaboration unit (embedded PC in our case). An added value of the proposed hardware is the absence of wires permitting the end-user to use a full range of motion during task execution without any restriction and allowing at the same time a quick patient preparation. Furthermore, it is possible to stream and record raw signal up to 6 h thanks to the rechargeable batteries.

Probes were placed on two muscles of the lower limbs that were involved throughout the execution of the activities to obtain raw EMG data. These muscles are commonly used during the execution of exercises that are typically employed to assess physical performance for sarcopenia evaluation. In other works, concerning the assessment of muscle behaviour during walking, good performance in literature was achieved by analysing the Gastrocnemius Lateralis and Tibialis Anterior muscles [16,28,29]. For this reason, also in this work these muscles were monitored. One probe was placed on each muscle on both legs; thus, a total of 4 probes and channels were considered. The probes are placed along the approximated direction of muscle fibres, with the inter-electrode distance of about 20mm to obtain the maximal surface EMG amplitude. The electrodes for Tibialis Anterior muscles are applied at about 1/3 of the distance between the tip of the fibula and the tip of the medial malleolus. As for Gastrocnemius, the electrodes are placed at about 1/3 of the line head of fibula on the most prominent bulge of the muscle.

From the software point of view, a real-time application has been implemented. The interface design is user-friendly so that it could also be used by medical professionals or caregivers. The main functions offered are: (1) display of the connection status of the probes (and relative battery life), (2) entry of the end-user fiscal code to associate the acquisition session with the user, (3) setting and pairing of the probes, (4) graphic display of the trend of the raw signals, (5) start and stop of acquisition for any sub-session, (6) buttons for feature processing and classification of sarcopenia confidence level, (7) visual label with indication of the confidence level.

The algorithmic pipeline designed for the acquisition, processing, elaboration of the raw EMG signal and classification of sarcopenia confidence level consists of three main blocks. In the first block, signal pre-processing techniques have been integrated (such as filtering and/or normalization of the signals) with the aim of formatting the data for the next step of extracting the features. The next block of the pipeline implements feature extraction methodologies, and it is followed by a further logic block designed for the selection of the most effective set of EMG features for evaluation of the considered pathology. Finally, a module for the classification of sarcopenia confidence level was implemented. Each block of the pipeline is detailed in the following. An overview of the platform with a block diagram representation is shown in Figure 3.

### 2.1. DATA Acquisition & Augmentation

A data collection was carried out to test the system’s performance. A total of 32 patients (19 males, with an average age of 63.95 ± 5.54 years old and 13 females, with an average age of 65.62 ± 7.30 years old) were recruited from Casa Sollievo della Sofferenza Hospital in San Giovanni Rotondo (Foggia, Italy). All patients were considered at risk or suffering from sarcopenia. A larger number of patients were initially planned for testing, but due to the COVID-19 emergency, the initial 6-month testing phase was reduced to about 2 months. Testing was performed considering: (a) the SARC-F questionnaire, (b) the muscle strength analysis through the hand grip-strength test and (c) the functional performance evaluation by means of sit-to-stand and gait speed tests.

The SARC-F may be considered a suitable tool for community screening for sarcopenia [30]. The questionnaire looks at the symptoms of sarcopenia that users have experienced, such as weakness, the need for assistance when walking, difficulty getting out of a chair, difficulties mounting stairs and falling. Each of the self-reported parameters is given a score between 0 and 2 for its minimum and maximum values. SARC-F values ≥ 4 are associated with the limitation of physical activities and risk of sarcopenia.

The hand grip-strength test was performed using a hand dynamometer with 2 trials for each hand and alternating sides during the test. The maximum values measured during all trials were considered for the analysis. As the criterion to define weak grip strength was considered the cut-offs suggested by the reports of the European Working Group on Sarcopenia in Older People (EWGSOP2) [12]: 27 kg for men and 16 kg for women.

Sit-to-stand and gait speed tests were performed as shown in Figure 4 and described in the following. Participants stood up as fast as possible from a sitting position to a standing position without help from their arms, which were held across their chest or extended to their sides. After, they walked 5 m and their average speed was measured. Participants wore sEMG sensors as described in the previous paragraph. The sit-to-stand phase was used to acquire electromyographic data useful for evaluating lower body strength, while gait speed was considered to analyse the physical performance of users. Slow gait speed was defined using EWGSOP2 reference value of <0.8 m/s [12].

To reduce the interindividual variability of EMG signals among different users, Maximum Voluntary Contraction (MVC) values are calculated. For MVC evaluation, the mean of data is estimated in the following three conditions: (1) the subject is in rest state for a period of 5 s to obtain a baseline signal; (2) the subject performs plantar flexion of the ankle against a fixed resistance and keeps it constant for 5 s to obtain the highest possible sEMG signal, resulting from the contraction of the Gastrocnemius Lateralis muscle; (3) the subject performs plantar flexion of the ankle against a fixed resistance and keeps it constant for 5 s to obtain the highest possible sEMG signal resulting from the contraction of the Tibialis Anterior muscle. The mean of the values thus acquired was used to normalize the processed data.

To define the confidence level of sarcopenia in the patients under examination and label the acquired electromyographic signals accordingly, the following criteria were applied:confidence level 1: if SARC-F ≥ 4;confidence level 2: if SARC-F ≥ 4 and hand grip-strength < cut-off values;confidence level 3: if SARC-F ≥ 4 and hand grip-strength < cut-off values and gait speed < 0.8 m/s.

The guidelines in “EWGSOP2” were considered to define the sarcopenia confidence levels, but the intrusive muscle mass assessment test was not done. The confidence levels were calculated using the SARC-F questionnaire, muscle strength testing and a physical performance evaluation (walking speed and sit-to-down test). Furthermore, it has been considered that some more recent research work [31,32] reveals a low level of confidence in the disease under investigation by only analysing the SARC-F questionnaire score.

As a result, 32 patients were separated into three groups: three with confidence level 1, twenty-two with confidence level 2, and seven with confidence level 3. An oversampling strategy was used to avoid unbalanced data, which is a prevalent problem in medicine. In particular, our dataset was pre-processed using a Synthesizing Minority Oversampling Technology (SMOTE) in combination with Edited Nearest Neighbours (ENN). SMOTE+ENN [33] is a sampling method, combining SMOTE [34] and Wilson’s ENN [35]. Specifically, SMOTE is an oversampling method, generating new minority class examples by interpolating between different minority class examples detected together. However, this method can also generate noise and boundary samples. Therefore, to obtain better-defined class clusters, ENN is used because it can remove any example whose class label differs from the class of at least two of its three nearest neighbours. Thus, it was shown that SMOTE+ENN reduces the potential overfitting in synthetic data.

### 2.2. Pre-Processing

The main steps of this phase are: (a) noise reduction, (b) EMG enveloping and (c) data normalization. The purpose of the first step is to reduce baseline noise and signal artifacts due to EMG electrode movements [36]. This was obtained by filtering the raw signals using a 4th order Butterworth band-pass filter with a frequency from 20 Hz to 450 Hz.

Subsequently, to make the signals comparable and suitable for further processing, the linear signal envelope was computed through full rectification and low-pass Butterworth filtering (with a cut-off frequency of 10 Hz). Finally, the normalization step was performed as described in Section 2.1.

### 2.3. Feature Extraction & Selection

The goal of the feature extraction phase is to extract relevant information from the surface EMG signal that can be used to identify muscle problems. Several time domain and time–frequency domain features utilized in medical and technical applications for monitoring lower-limb muscles were investigated for this study [37,38,39,40]. The main features that have been investigated are shown in Table 1.

In this study, the size of the sliding window was set to 200 ms, with an incremental window to 50 ms [41]. After segmenting the EMG data, twelve EMG features were extracted from each EMG channel, so the feature dimensional vector is 48. To reduce the complexity of the signal processing and increase the performance of the system, the Modified Binary Tree Growth Algorithm (MBTGA) feature selection technique was applied to select the most effective EMG feature subset. The MBTGA was adopted since it showed good performance for the analysis of EMG signals [42].

The MBTGA is an optimization of the Binary Tree Growth Algorithm (BTGA), developed to enhance performance in EMG feature selection.

The best subset of features, selected through the MBTGA, consists of the following three features: Integrated EMG, Root Mean Square, Averaged Instantaneous Frequency for all channels; so, the dimension of the feature vector is 12.

### 2.4. Classification

After feature extraction and selection, the data were labelled as different classes according to the sarcopenia confidence level as previously described. Eight ML algorithms were trained on this data for comparison: SVM, Decision Tree (DT), Random Forest (RF), Logistic Regression (LR), K-Nearest Neighbors (KNN), Naïve Bayes (NB), Multi-layer Perceptron (MLP) and Extreme Gradient Boosting (XGB).

SVM [43] is a classification and regression method developed in the context of statistical learning theory. It has been shown to perform better in terms of accuracy than other classifiers in different application domains, and, in addition, to be efficiently scalable for large problems. SVM attempts to find a hyperplane in N-dimensional space (where N is the number of features) distinguishing the data points so that the margin or distance between each data set and the baseline for classifying the data is maximized. It identifies a set of examples, called support vectors, which appear to be the most representative observations for each target class. A kernel is used to implement SVM algorithms. Most commonly used kernels are the linear kernel, polynomial kernel, and radial kernel. In our approach a linear kernel was applied.

DT is a popular supervised ML algorithms [44]. Specifically, in DT, data are split according to a certain parameter. In this algorithm, a tree is used as a predictive model in order to traverse the observations about a feature (represented by the branches of the tree) and to arrive at the feature’s target value (represented by the leaves); in particular, the leaves represent the class labels and the branches represent the feature conjunctions resulting in the class labels. In our tests the maximum tree depth is fixed to 10.

RF algorithm, introduced by [45], constructs a set of predictors with a set of decision trees that are randomly generated in datasets. It uses almost the hyper-parameters of a decision tree. In particular, in order to classify the input vector, each classifier is generated using a vector that is independent of the input vector, and each tree votes for the largest number of classes. RF adds more randomness to the model while increasing the trees. It detects the best feature in a random subset of features. In our approach the number of estimators in the forest is fixed to 10, whereas the maximum tree depth is set to 10.

LR is a statistical method for converting binary classification problems into linear regression ones. LR classifies values by applying a standard logistic function, known as sigmoid functionand it can be found in many applications of biological, economic and statistic fields [46]. In multiclass classification, such as the three classes in our approach, LR applies the one versus the rest method. In detail, this method generates and trains LR models for each class compared to the rest of the classes.

KNN is a popular classification method due to its easy implementation and high classification performance. However, the idea of the algorithm is the assignment of a sample to a category if most of the k nearest neighbor samples of the considered samples belong to the same category. Usually, k is not greater than 20 [47]. The choice of k is important because: if k is too small, the approach is sensitive to noise, while if k is too large, the neighborhood may include samples from other classes. The selected neighbors are those that have been correctly classified.

NB classifier is a popular algorithm due to its simplicity and linear run-time [48]. It assumes that variables are independent of the given classes and that numeric attribute values are normally distributed within each class, but in many real-world datasets the latter condition is strongly violated.

MLP algorithm [49] uses back-propagation where input data are continuously transmitted to the neural network and the output is compared with the desired output, allowing to estimate the error. Another step is a feedback process, where the error is returned to the neural network. The algorithm is efficient if, in each iteration, the output of the considered neural model is close to the desired output.

XGB [50] is designed to compensate for the drawbacks of gradient boosting. This algorithm allows fast classification and very good prediction results. It also allows overfitting regularization by internal cross-validation in each iterative step.

In Table 2 the optimal selected parameters for each ML model obtained through a grid search technique [51] are shown.

## 3. Results and Discussion

To validate the proposed pipeline, a series of experiments were performed to verify the effectiveness of the described approach and its operation in real-time. The dataset described in Section 2.1, containing patients with three confidence levels of sarcopenia (1, 2 and 3), was considered. Our EMG signal acquisition and processing system was developed through the following steps: (1) raw signal acquisition through software routines implemented in C#, (2) dataset balancing, data processing, feature extraction and classification using Python language (3.7.1). Our experiments were performed on an embedded PC with the Intel Core i5 processor and 8 GB of RAM. The classifiers’ performance has been evaluated using four different metrics: accuracy, precision, recall, F1-score. For these metrics, some terms are introduced in Table 3.

These metrics are defined by the following expressions:(1)$$\begin{array}{ccc} \mathrm{Acc} & = & \frac{\mathrm{TP}+\mathrm{TN}}{\mathrm{TP}+\mathrm{TN}+\mathrm{FP}+\mathrm{FN}} \end{array}$$(2)$$\begin{array}{ccc} \mathrm{Pr} & = & \frac{\mathrm{TP}}{\mathrm{TP}+\mathrm{FP}} \end{array}$$(3)Re=TPTP+FN(4)$$\begin{array}{ccc} F1-\mathrm{score} & = & \frac{2*\mathrm{TP}}{2*\mathrm{TN}+\mathrm{FP}+\mathrm{FN}} \end{array}$$

In particular, accuracy is the ratio between all correctly classified samples and all samples. Precision indicates how accurately the model predicts positive occurrences. Recall shows how the model is able to detect positive cases using all positive cases. F1-score has a greater impact on true positive cases than precision.

Since in our study, a multiclass classification problem is discussed, metrics such as accuracy, precision/recall or F1-score do not provide a complete overview of the compared classifiers’ performance. So, as reported in literature [52], Cohen’s Kappa is another essential performance indicator. In particular, Cohen’s Kappa is used to measure the agreement between the instance’s true label and the one predicted by the selected classifier. It is defined as:(5)$$k=\frac{p_{o}-p_{e}}{1-p_{e}}$$
where $p_{o}$ is the observed label and $p_{e}$ is the expected label. Cohen’s Kappa is always less than or equal to 1. In Table 4 the correspondance between Cohen’s Kappa and agreement is reported.

It is important to highlight that the analysed data set is unbalanced, particularly between the class with sarcopenia confidence level 2 and the other two. In particular it can be observed that the ratio of confidence level 1 versus confidence level 2 is 1:7.33. So, to avoid this unbalance and to generate a more robust dataset, an augmentation strategy was used, as described in Section 2.1, resulting in a dataset of 150 patients with balanced sarcopenia confidence levels (47–52–51).

In this work, each ML model was trained by first considering all available features and then considering only features obtained by the feature selection technique described in Section 2.3. Subsequently, the performances of such models were compared on the basis of separately constructed test sets. A 10-cross-validation [53] was applied. In order to reduce classification bias, this procedure is used to perturb the training set of each classifier randomising the original data set. So, each classifier is trained for each fold using 90% of the data, whereas the remaining 10% is used for testing. In addition, to avoid over-fitting of the training set, 10% of the training data is additionally used to create a validation set. The procedure is repeated 10 times training the classifier with a different training set and testing with a separated test set. It is important that the same samples do not appear in the training and test sets at the same time.

Table 5 and Table 6 present the performance of each ML model without feature selection and with feature selection, respectively. SVM model showed the best performance in terms of accuracy, precision, recall and F1-score both considering all features and those obtained via feature selection. On the other hand, the NB model showed the worst performance. A significant improvement in model performance can be seen by applying feature selection with a range of 10–20% improvement. In particular, SVM has a 10% improvement in accuracy from 86.7% to 96.7%, while DT and NB have a 20% improvement respectively from 66.7% to 86.7% and from 62.8% to 82.8%.

Table 7 shows the Cohen’s Kappa values for the various compared classifiers both without and with feature selection. The numerical values, shown in the first line of the table, highlight the high sensitivity of the considered metric. We can see that 5 classifiers give a moderate agreement and the remaining 3 classifiers provide a good agreement. Instead, with the feature selection, the range of the obtained values varying the classifiers is more limited, with a gap between the best and the worst classifier of about 24.5%. Additionally, for this metric, SVM is the classifier with the best score among those that obtain a perfect agreement (LR, KNN, MLP and XGB). Finally, the differences in Cohen’s Kappa values for each classifier confirm the goodness of the introduction of the feature selection logical block in the proposed algorithmic pipeline.

In a multi-class recognition problem, as in the present study, the use of an average recognition rate (i.e., accuracy) among all the classes could not be exhaustive due to the impossibility of inspecting the separation level in terms of correct classifications, among classes (in our case, the three different sarcopenia confidence levels). To overcome this limitation, in Figure 5 and Figure 6 the confusion matrices of the average accuracies obtained for each considered classifier with and without feature selection are reported. From the confusion matrices analysis, it can be seen that for the classifier with the best performance, there are failures to classify classes adjacent to each other; in particular, SVM confuses confidence level 2 with the other two.

## 4. Conclusions

In recent years, the “intelligent healthcare” concept has become more and more consistent, due to the commercialization of smart medical devices. Moreover, the use of artificial intelligence to build clinical DSS has achieved excellent results, especially with regard to the identification and/or prevention of specific diseases.

The primary objectives of this work are essentially two. Firstly, an algorithmic pipeline was designed and implemented with the aim of classifying three different confidence levels of sarcopenia using electromyographic signals acquired through a commercial device (sEMG). Secondly, this research provides a wide overview of the relative performance of different supervised ML algorithms for assessing the severity level of the disease. Due to the pandemic period and the consequent difficulty in obtaining real data, algorithmic approaches to produce synthetic data from the electromyographic signals of only 32 subjects was implemented.

The obtained results confirmed the algorithmic choices made, in fact all the measured metrics showed a numerical improvement using the “feature selection” logic block. SVM classifier outperformed the other 8 supervised classifiers. However, it is necessary to underline an important limitation of the presented study. To compare the performance of the classification algorithms, only synthesized data from a specific technique (even if consolidated in the scientific literature) are considered.

Future work will include the implementation of other ML algorithms with a larger dataset and also considering image-based datasets. Through the latter type of data, it will be possible to apply Deep Learning algorithms for the classification of different confidence levels of sarcopenia. Further future development of this work will consist of applying different feature selection techniques and varying the clinical protocol. In particular, a new testing protocol will be evaluated to be able to assess muscle behaviour, for the diagnosis of sarcopenia, even for subjects who are unable to perform sitting-to-standing and gait speed tests, for example due to injury or gait problems. Last but not least, other muscles (not only those of the lower limbs) will be analysed and long-term monitoring of electromyographic signals will be carried out to create an intelligent and automatic tool for the early diagnosis of the considered pathology.

## Figures and Tables

**Figure 1 sensors-22-02721-f001:**
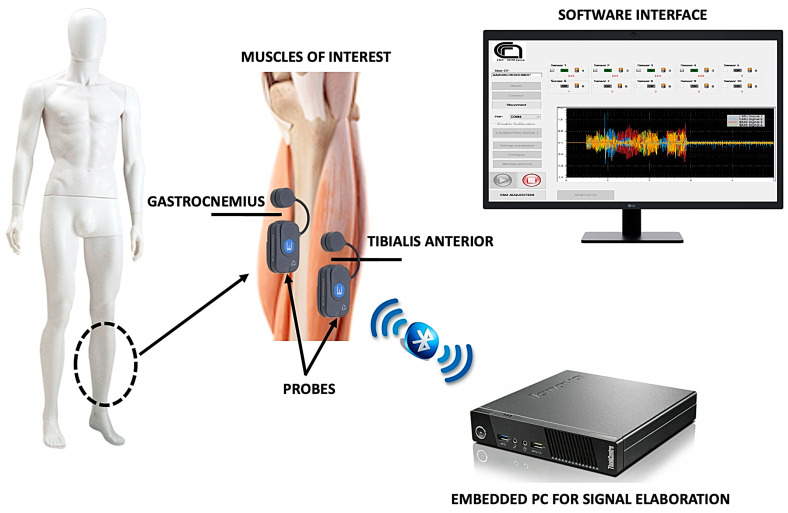
Schematic overview of the proposed sEMG platform.

**Figure 2 sensors-22-02721-f002:**
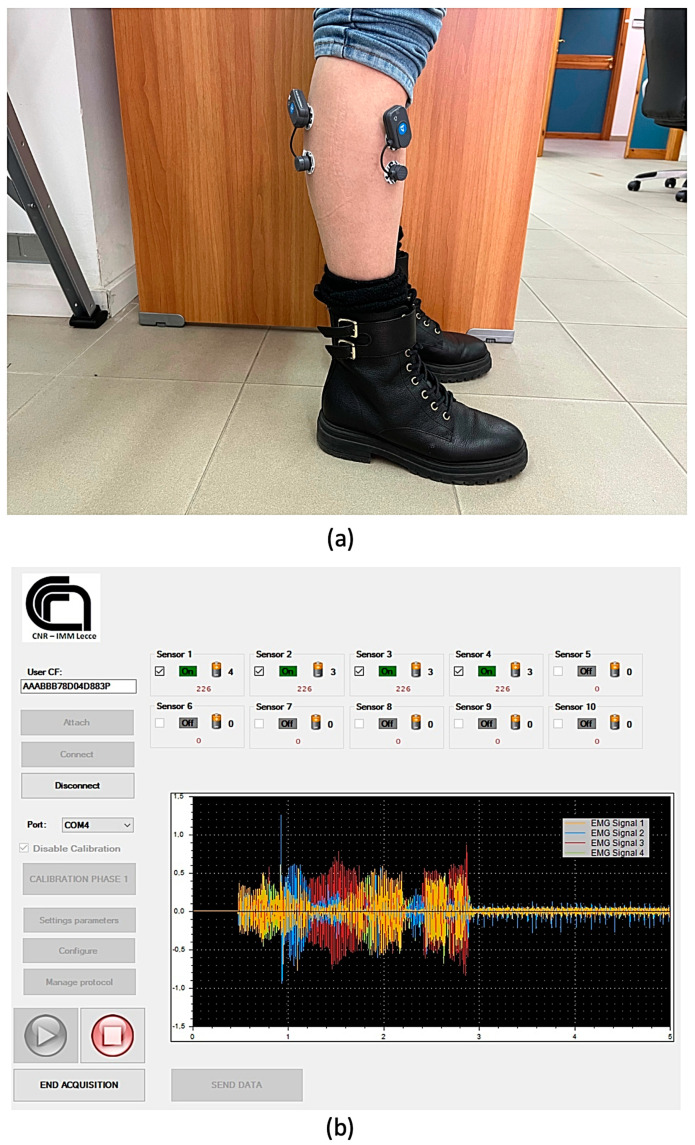
(**a**) Positioning of the electrodes on Gastrocnemius Lateralis and Tibialis Anterior muscles; (**b**) software interface developed for raw sEMG data acquisition and elaboration.

**Figure 3 sensors-22-02721-f003:**
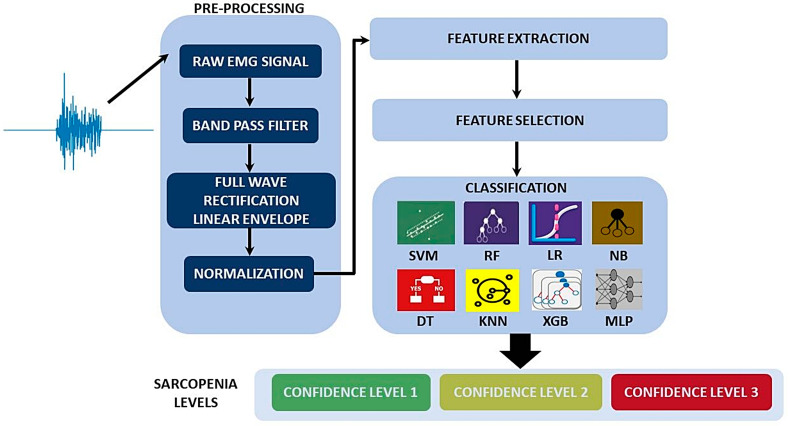
Overview of the proposed algorithmic pipeline designed and implemented for the distinction of different confidence levels of sarcopenia. The pipeline reports within the classification block the supervised classifiers compared in our work.

**Figure 4 sensors-22-02721-f004:**
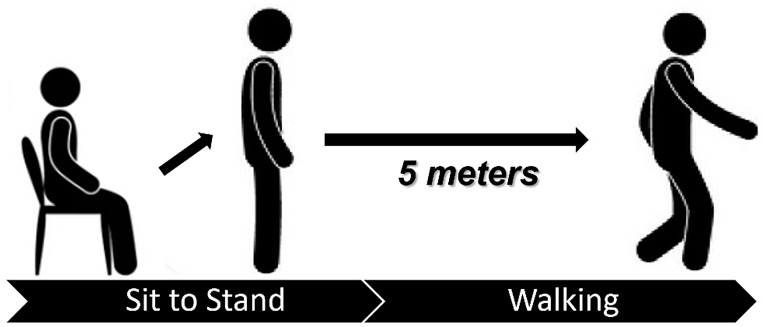
Sit-to-stand and gait speed tests.

**Figure 5 sensors-22-02721-f005:**
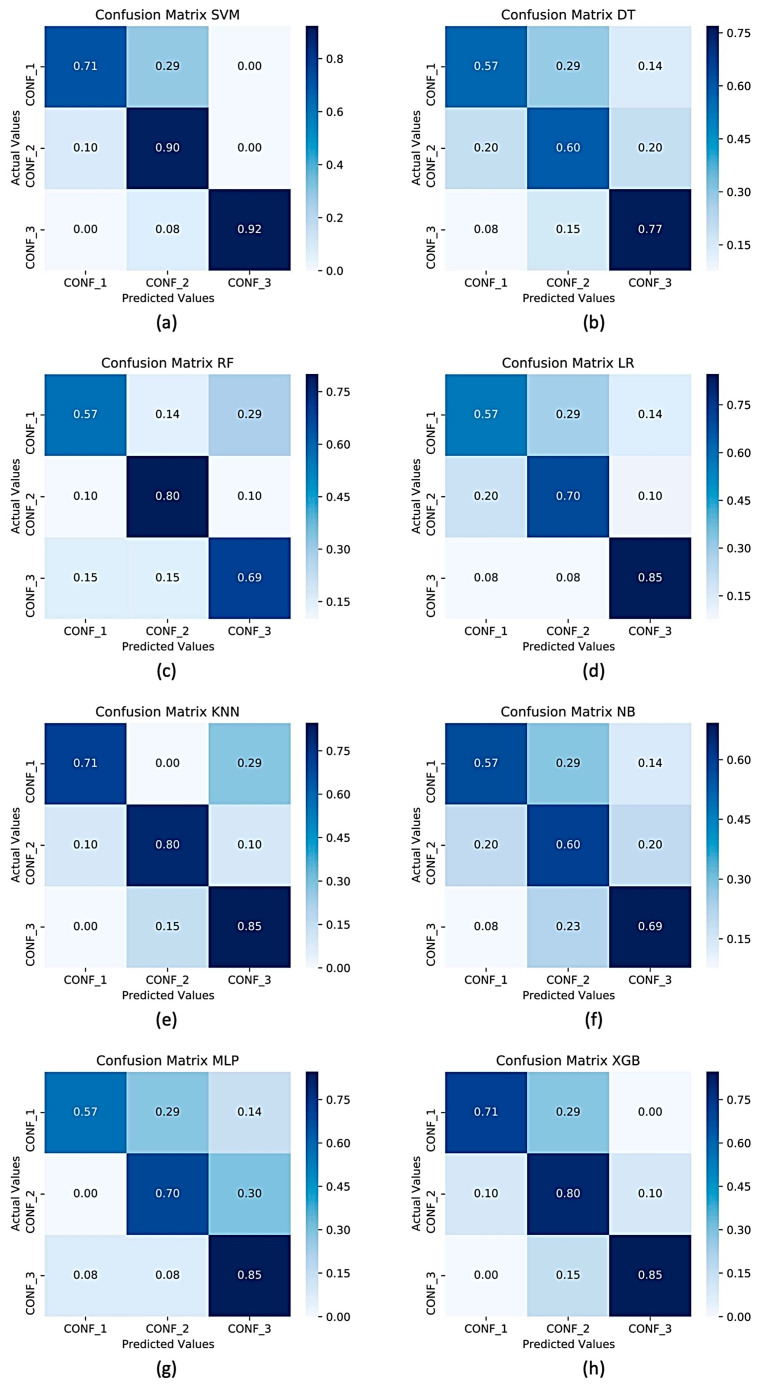
Confusion matrices for three classes of sarcopenia confidence levels using (**a**) SVM, (**b**) DT, (**c**) RF, (**d**) LR, (**e**) KNN, (**f**) NB, (**g**) MLP, (**h**) XGB as classifiers without feature selection.

**Figure 6 sensors-22-02721-f006:**
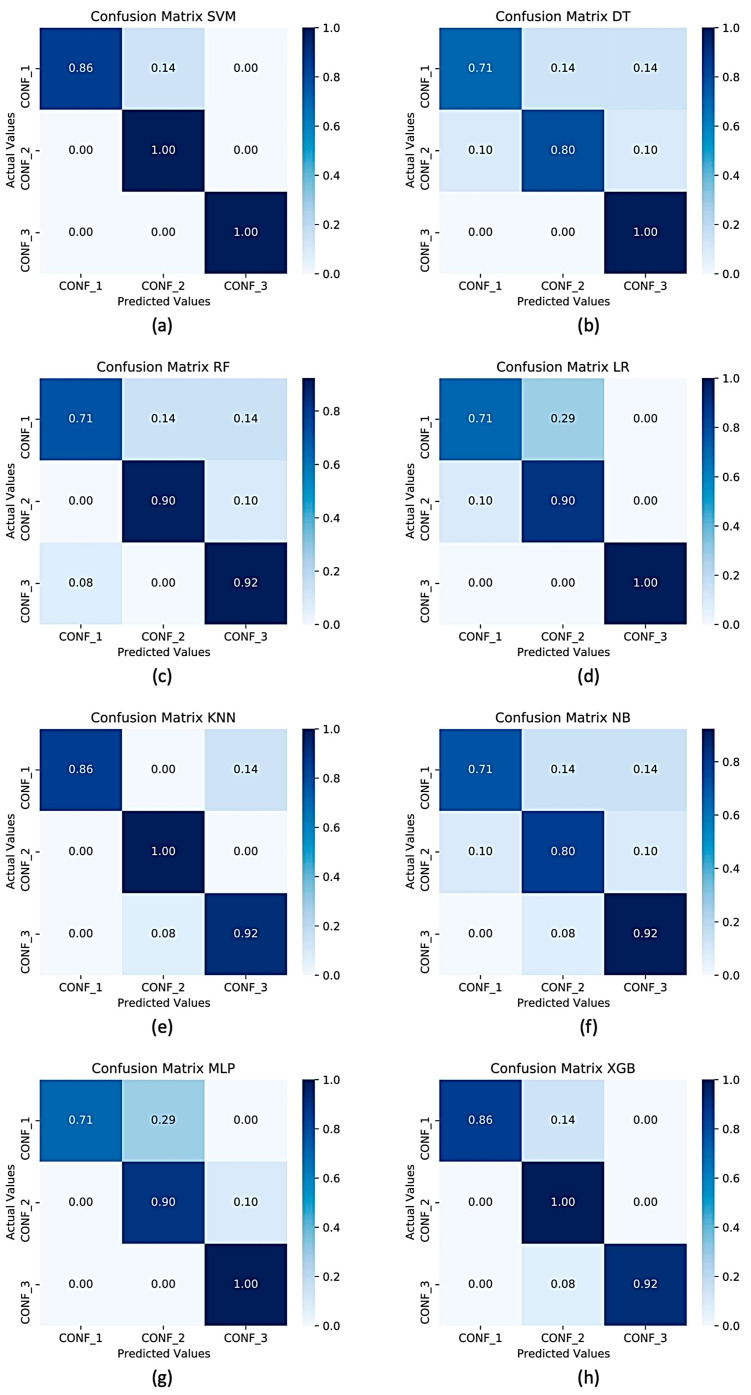
Confusion matrices for three classes of sarcopenia confidence levels using (**a**) SVM, (**b**) DT, (**c**) RF, (**d**) LR, (**e**) KNN, (**f**) NB, (**g**) MLP, (**h**) XGB as classifiers with feature selection.

**Table 1 sensors-22-02721-t001:** Mathematical equations of the tested features for an EMG signal segment of length *N*.

Features Name	Formula
Integrated EMG (IEMG)	$IEMG=\sum_{i=1}^{N}\left|EMG_{i}\right|$
Mean absolute value (MAV)	$MAV=\frac{\sum_{i=1}^{N}\left|EMG_{i}\right|}{N}$
Modified mean absolute value type 1 (MAV1)	$\begin{array}{cc} MAV1=\frac{\sum_{i=1}^{N}k_{i}\left|EMG_{i}\right|}{N}\\ \; & k_{i}=\left\{\begin{array}{cc} 1, & \mathrm{if}\;0.25N\leq i\leq 0.75N\\ 0.5,\; & \mathrm{otherwise} \end{array}\right. \end{array}$
Modified mean absolute value type 2 (MAV2)	$\begin{array}{cc} MAV2=\frac{\sum_{i=1}^{N}k_{i}\left|EMG_{i}\right|}{N}\\ \; & k_{i}=\left\{\begin{array}{cc} 1, & \mathrm{if}\;0.25N\leq i\leq 0.75N\\ \frac{4i}{N}, & \mathrm{if}\;i<0.25N\\ \frac{4(i-N)}{N}, & \mathrm{otherwise} \end{array}\right. \end{array}$
Root Mean Square (RMS)	$RMS=\sqrt{\frac{\sum_{i=1}^{N}EMG_{i}^{2}}{N}}$
Variance (VAR)	$VAR=\frac{\sum_{i=1}^{N}k_{i}EMG_{i}^{2}}{N-1}$
Average amplitude change (AAC)	$AAC=\sum_{i=1}^{N-1}\left|EMG_{i+1}-EMG_{i}\right|$
Zero Crossing (ZC)	$\begin{array}{cc} ZC=\sum_{i=1}^{N-1}\left[sgn\left(EMG_{i}\times EMG_{i+1}\right)\cap \left|EMG_{i+1}-EMG_{i}\right|\geq 0\right]\\ \; & sgn\left(EMG\right)=\left\{\begin{array}{cc} 1, & \mathrm{if}\;EMG\geq thr\\ 0, & \mathrm{if}\;EMG<thr \end{array}\right.\\ \; & \mathrm{where threshold thr = 0.1 mV} \end{array}$
Simple Square Integral (SSI)	$SSI=\sum_{i=1}^{N}{\left|EMG_{i}\right|}^{2}$
Slope Sign Change (SSC)	$\begin{array}{cc} SSC=\sum_{i=2}^{N}\left[f\left[\left(EMG_{i}-EMG_{i-1}\right)\times \left(EMG_{i}-EMG_{i+1}\right)\right]\right]\\ \; & f\left(EMG\right)=\left\{\begin{array}{cc} 1, & \mathrm{if}\;EMG\geq thr\\ 0, & \mathrm{if}\;EMG<thr \end{array}\right.\\ \; & \mathrm{where threshold thr = 0.1mV} \end{array}$
Willison Amplitude (WAMP)	$\begin{array}{cc} WAMP=\sum_{i=1}^{N}f\left(\left|EMG_{i}-EMG_{i+1}\right|\right)\\ \; & f\left(EMG\right)=\left\{\begin{array}{cc} 1, & \mathrm{if}\;EMG\geq thr\\ 0, & \mathrm{if}\;EMG<thr \end{array}\right.\\ \; & \mathrm{where threshold thr = 0.1 mV} \end{array}$
Averaged Istantaneous Frequency (AIF)	$\begin{array}{cc} AIF=\frac{1}{t_{b}-t_{a}}\int_{t_{a}}^{t_{b}}w\left(t\right)\;dt\\ \; & [t_{b}t_{a}]\;\mathrm{time window of calculation}\\ \; & w(t)\;\mathrm{is the instantaneous frequency of the signal} \end{array}$

**Table 2 sensors-22-02721-t002:** Parameters used for classification models.

Model	Parameters
SVM	decision_function_shape = ovo, max_iter = 100, kernel = linear, C = 0.1
DT	criterion = gini, max_depth = 10
RF	max_depth = 10, n_estimators = 10, criterion = gini
LR	solver = newton-cg, max_iter = 50, multi_class = ovr, C = 0.001
KNN	n_neighbors = 5, metric = minkowski, algorithms = auto, weights = distance
NB	var_smoothing = 0.00001
MLP	activation = identity, alpha = 0.0001, hidden_layer_sizes = (10, 10), solver = lbfgs
XGB	learning_rate = 0.001, max_depth = 2, n_estimators = 214

**Table 3 sensors-22-02721-t003:** Definition of the terms used in metrics.

Predicted Label	Actual Label	Definition
Positive	Positive	True Positive (TP)
Positive	Negative	False Positive (FP)
Negative	Positive	False Negative (FN)
Negative	Negative	True Negative (TN)

**Table 4 sensors-22-02721-t004:** Cohen’s Kappa vs agreement.

Cohen’s Kappa	Agreement
k<0.20	slight
0.21≤k<0.40	fair
0.41≤k<0.60	moderate
0.61≤k<0.80	good
0.81≤k≤1.00	perfect

**Table 5 sensors-22-02721-t005:** Classifier results with dataset augmentation and without feature selection.

Model	Accuracy	Precision	Recall	F1
SVM	0.867	0.861	0.845	0.849
DT	0.667	0.647	0.647	0.647
RF	0.698	0.683	0.688	0.684
LR	0.728	0.706	0.706	0.706
KNN	0.797	0.806	0.787	0.797
NB	0.628	0.622	0.621	0.621
MLP	0.729	0.744	0.706	0.717
XGB	0.798	0.806	0.787	0.798

**Table 6 sensors-22-02721-t006:** Classifier results with dataset augmentation and feature selection.

Model	Accuracy	Precision	Recall	F1
SVM	0.967	0.969	0.953	0.956
DT	0.867	0.862	0.838	0.847
RF	0.867	0.863	0.845	0.852
LR	0.9	0.884	0.871	0.875
KNN	0.933	0.941	0.926	0.932
NB	0.828	0.830	0.812	0.819
MLP	0.9	0.916	0.871	0.884
XGB	0.936	0.944	0.927	0.931

**Table 7 sensors-22-02721-t007:** Cohen’s Kappa results without and with feature selection.

	SVM	DT	RF	LR	KNN	NB	MLP	XGB
without FS	0.794	0.485	0.538	0.588	0.687	0.436	0.579	0.691
with FS	0.984	0.791	0.792	0.845	0.896	0.739	0.843	0.897

## Data Availability

The data presented in this study are available on request from the corresponding author. The data are not publicly available due to restrictions (they contain information that could compromise the privacy of research participants).

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
