# Peer review of "Comparative Analysis of Supervised Classifiers for the Evaluation of Sarcopenia Using a sEMG-Based Platform"

_sensors, 2022, doi:10.3390/s22072721_

Round 1

Reviewer 1 Report

Ciao, complimenti per la vostra ricerca.

It is a very interesting and helpful study; however, some points need adjustments.

Abstract

P1 L3 “To reduce the effects of the disease, it is important to recognize the level and progression of sarcopenia early.”

  1. To reduce the effects of the disease, just the diagnostic is not enough, it is necessary more than recognition. Consider been less emphatic.

Introduction

Good characterization and association of sarcopenia and its economic impact.

P2 L41 “The diagnosis of sarcopenia is very difficult.”

  1. Consider been less emphatic.

P2 L60-65

  1. Consider summarizing the paragraph.

P3 L126-30

  1. You can exclude this paragraph without jeopardizing the subject.

P4

  1. Figure 1: Improve the muscle localization, and the position of the electrodes, too. It is important a figure showing the experimental setup with the subject positioning and equipment. It is not mandatory to do it in the same figure, feel free to build a new one.

P4 L164

  1. It was used reference electrode?

P5 L179

  1. How was the criterion to normalize the data? It is important to show it in the manuscript.

P6 L217

  1. It is not clear how was obtained the MVC, were they standing or sited?
  2. Please, justify why it was not measured the strength, once it was well described in the introduction that sarcopenia is strictly associated with the loss of strength.

Conclusions

It is well described showing the main findings, limitations, and future perspectives.

Author Response

QUESTION
P1 L3 “To reduce the effects of the disease, it is important to recognize the level and progression of sarcopenia early.”
To reduce the effects of the disease, just the diagnostic is not enough, it is necessary more than recognition. Consider been less emphatic.
RESPONSE
- We thank the reviewer for the observation. In the revised version of the manuscript the sentence has been re-edited according to the provided suggestion.

QUESTION
Introduction
Good characterization and association of sarcopenia and its economic impact.
P2 L41 “The diagnosis of sarcopenia is very difficult.” Consider been less emphatic.
RESPONSE
- We thank the reviewer for the suggestion. We have re-edit the sentence.

QUESTION
P2 L60-65
Consider summarizing the paragraph.
RESPONSE
- In the revised version of the manuscript the entire paragraph was re-edited.

QUESTION
P3 L126-30
You can exclude this paragraph without jeopardizing the subject.
RESPONSE
- Thank you for your kind comment. In the revised version of the manuscript the paragraph was deleted.

QUESTION
P4
Figure 1: Improve the muscle localization, and the position of the electrodes, too. It is important a figure showing the experimental setup with the subject positioning and equipment. It is not mandatory to do it in the same figure, feel free to build a new one.

RESPONSE - We thank the reviewer. In the revised version of the manuscript a new figure (figure 2) has been added to detail the positioning of the sensors on the target muscles and to show how the implemented software interface works.

QUESTION
P4 L164
It was used reference electrode?
RESPONSE - No, the probes of BTS FREEEMG1000 platform use two active electrodes, without reference electrode.

QUESTION
P5 L179
How was the criterion to normalize the data? It is important to show it in the manuscript.
RESPONSE
- We thank the reviewer for this remark. MVC normalization was accomplished. To clarify this aspect, some sentences have been added in the paper.

QUESTION
P6 L217
It is not clear how was obtained the MVC, were they standing or sited?
Please, justify why it was not measured the strength, once it was well described in the introduction that sarcopenia is strictly associated with the loss of strength.
RESPONSE
- We thank the reviewer. To obtain the MVC the users were in sitting position. Some sentences have been added in the manuscript to clarify this point. Moreover, EMG features correlated with strength and muscle behaviour were extracted and analysed.

QUESTION
Conclusions
It is well described showing the main findings, limitations, and future perspectives.
RESPONSE
- We thank the reviewer for appreciating the organization of the "Conclusions" section

Reviewer 2 Report

Sarcopenia deserves to be studied in greater detail, and this research is to be welcomed.  However, I have difficulty relating it to sensors.  Aim 1 is to design and implementation of a hardware/software platform. Aim 2 is to compare the efficiency of classifiers in the evaluation of sarcopenia.  Neither of these suggest that sensors have a research aim.

The conclusion states that electromyographic signals were acquired through a commercial device.  Gel electrodes were used to obtain the raw signals, captured by bipolar surface electrodes placed on the skin. There is no indication that any of this relates to research objectives – indeed, it assumes the research on data acquisition has already been done.

If I were to make any recommendation to enhance the paper, it would be to provide more basic data on the signals derived from the gel sensors, and to explain how the authors justify using this data to assess sarcopenia. 

The work on the hardware/software platform and the analysis of classifiers appears to me to be useful and rigorous.

Typos: 104: trough            278 perfermance

English language: 205: splitted (Split is an irregular verb.  The past tense is split, not splitted.)

Author Response

QUESTION
Sarcopenia deserves to be studied in greater detail, and this research is to be welcomed. However, I have difficulty relating it to sensors. Aim 1 is to design and implementation of a hardware/software platform. Aim 2 is to compare the efficiency of classifiers in the evaluation of sarcopenia. Neither of these suggest that sensors have a research aim. The conclusion states that electromyographic signals were acquired through a commercial device. Gel electrodes were used to obtain the raw signals, captured by bipolar surface electrodes placed on the skin. There is no indication that any of this relates to research objectives – indeed,
it assumes the research on data acquisition has already been done. If I were to make any recommendation to enhance the paper, it would be to provide more basic data on the signals derived from the gel sensors, and to explain how the authors justify using this data to assess sarcopenia.
RESPONSE
- We thank the reviewer. The design and implementation of the hardware/software platform focused first on the identification and testing of a commercial EMG device that could be suitable for developing an easy-to-use Decision Support System (DSS) that can also be used in nursing homes (or home settings). So, the focus has been on the selection and testing of sensors that would be minimally invasive, wireless, certified, easily manageable, stable and allowing open access to raw data. After that, the maximum effort was addressed principally for the following tasks: 1) the development of the software framework suitable to process the data coming from the chosen hardware, 2) the extraction of useful information for sarcopenia evaluation, applying machine
learning techniques. Moreover, the raw sEMG signal is noisy and not easy to interpret visually.
Consequently, useful information for the classification of different sarcopenia levels can be obtained only by processing the signal (filtering noise, artifact reduction, normalization, etc.) and extracting mathematical features given as input to a machine learning classifier.

QUESTION
The work on the hardware/software platform and the analysis of classifiers appears to me to be useful and rigorous.
Typos: 104: trough 278 perfermance
English language: 205: splitted (Split is an irregular verb. The past tense is split, not splitted.)
RESPONSE
- We thank the reviewer. In the revised version of the manuscript the suggested changes have been made

Round 2

Reviewer 1 Report

All my concerns were answered.